# “Get Well Enough to Make the Right Decision for Themselves”—Experiences and Perspectives of Clinicians Working with People with Serious Mental Illness and Their Substitute Decision Makers

**DOI:** 10.3390/bs15050704

**Published:** 2025-05-20

**Authors:** Samuel Law, Vicky Stergiopoulos, Juveria Zaheer, Arash Nakhost

**Affiliations:** 1Department of Psychiatry, Faculty of Medicine, University of Toronto, Toronto, ON M5T 1R8, Canada; vicky.stergiopoulos@camh.ca (V.S.); juveria.zaheer@camh.ca (J.Z.); 2MAP Centre for Urban Health Solutions, Li Ka Shing Knowledge Institute, St. Michael’s Hospital, Unity Health Toronto, Toronto, ON M5C 2T2, Canada; 3Department of Psychiatry, St. Michael’s Hospital, Unity Health Toronto, Toronto, ON M5C 2T2, Canada; 4Centre for Addictions and Mental Health, Toronto, ON M6J 1H4, Canada; 5Department of Psychiatry, McGill University, Montreal, QC H3A 1A1, Canada; arash.nakhost@mcgill.ca

**Keywords:** substitute decision making, serious mental illness, family, medical–legal, Convention on the Rights of Persons with Disability (CRPD)

## Abstract

In the current clinical psychiatric practice in most of the world, treatment decisions are based on a person’s capacity to make these decisions. When a person lacks the capacity to understand and appreciate treatment decisions, in many jurisdictions a third-party substitute decision maker (SDM) is appointed on his or her behalf in order to promote safety and optimal clinical outcome. In Ontario, Canada, for example, family members (typically) or public guardians are appointed as SDMs, and they form an integral part of the medical–legal system in psychiatric care. Clinicians working with both patients and their SDMs in these circumstances encounter unique challenges and deliver care in specialized ways, though little research has focused on their experiences and reflections. Based on focus group data, this qualitative study uses a descriptive and interpretative phenomenological approach through thematic analysis to examine these aspects from clinicians working in both inpatient and outpatient settings of an urban teaching hospital’s psychiatric services in Toronto, Canada. Seven key themes emerged: Clinicians (1) appreciate hardships and challenges in lives of SDMs and patients—including the challenging emotions and experiences on both sides, and the risks and relational changes from being an SDM; (2) have an understanding of the patient’s situation and respect for patient autonomy and wishes—they are promoter of autonomy and mindful of patients’ prior wishes amidst patients’ fluctuating capacity, facilitating communication, keeping patients informed and promoting transitioning from SDM to self-determination; (3) have a special working relationship with family SDMs—including supporting SDMs, avoiding harm from delayed or denied treatment, and educating and collaborating with SDMs while maintaining professional boundaries; (4) at times find it difficult working with SDMs—stemming from working with over-involved or uninterested family SDMs, coping with perceived poor SDM decisions, and they sometimes ponder if SDMs are necessary; (5) delineate differences between family and Public Guardian and Trustee (PGT) SDMs—they see PGT as closely aligned with medical decision makers, while family SDMs are more intimately involved and more likely to disagree with a physician’s recommendation; (6) recognize the importance of the SDM role in various contexts—through seeing social values in having SDMs, and acknowledging that having SDMS help them to feel better about their actions as they work to protect the patients; and (7) express ideas on how to improve the current system—at public, societal, and family SDM levels. We conclude that clinicians have unique mediating roles, with privilege and responsibility in understanding the different roles and challenges patients and SDMs face, and have opportunities to improve patient and SDM experiences, clinical outcomes, carry out education, and advocate for ethically just decisions. These clinical roles also come with frustration, discomfort, moral distress and at times vicarious trauma. Clinicians’ unique understanding of this complex and nuanced intersection of patient care provides insight into the core issues of autonomy, duty to care and protect, advocacy, and emotional dynamics involved in this sector as a larger philosophical and social movement to abolish SDMs, as advocated by the Convention on the Rights of Persons with Disability (CRPD), is taking place. We briefly discuss the role of supported decision making as an alternative as.

## 1. Introduction

The role of clinicians in caring for individuals who lack the capacity to make treatment decisions is complex and ethically challenging. Individuals of majority age are entitled to the right to make their own treatment decisions, with presumed commensurate cognitive or mental capacity to make an informed decision ([83]). In many cases, when a person is found to lack such capacity to make informed treatment decisions, a substitute decision maker (SDM) is appointed to act on his or her behalf. The processes used to appoint and enact substitute decision making differ across jurisdictions, but the essential aim is to have a third party—e.g., a family member, a power of attorney, a court appointed person or clinician, a public guardian, or a mental health tribunal—to help those without the capacity to make decisions that will serve their health and well-being interests ([16]; [79]).

In clinical practice, substituted decision making often encompasses and co-exists on a continuum with ‘supported decision-making’ ([17]). On the one end, substituted decision making refers to decisions that are made on a person’s behalf by a third party in his or her objective medical best interests. On the other end, supported decision making refers to situations where the decision is made by the patient after being offered informal and formal support, and if after substantial efforts have been made a person is still unable to make a clear treatment decision, then a decision will be made for him or her by a third party who will base it on the best interpretation of his or her will and preferences. That is, a substituted judgment model ([1]). In many developed countries including Canada, if and when such treatment decision-making ability is in question, it is the responsibility of a regulated healthcare professional—typically a physician—to formally assess the patient’s decisional capacity and identify an SDM if necessary.

In Ontario, Canada, for example, the capacity assessment is based on a standard approach ([31]) that evaluates an individual’s ability to understand the relevant information related to a decision and appreciate the foreseeable consequences of a decision, or lack thereof ([53]). When individuals are deemed incapable, SDMs are appointed to make such decisions on their behalf. In this jurisdiction, the SDM may be a legal guardian, an appointed power of attorney, a family member or relative from a ranked list based on proximity of kinship, or, when none of the above is available or willing, a Public Guardian and Trustee (PGT) ([53]). SDMs’ decision making is guided by a set of criteria defined by the accepted practices of the jurisdiction—in Ontario’s case, it is to consider a patient’s last capable wishes (i.e., not necessarily what is perceived by the SDM to be appropriate, wise, or in the patient’s current best interest) ([30]). In some jurisdictions, the Ontario SDM may be more akin to that of a guardian ([80]; [50]).

While formally the appointed SDM has the legal right to make treatment decisions, SDMs are usually encouraged and expected to use supported decision making to help patients who may have partial mental capacity to make a more autonomous decision. Furthermore, SDMs are also better off if they consult their patients, even if they have lost mental capacity, and take into account their views to the extent they are able to do so, and have them included in the process as much as possible ([78]).

The matter of substitute decision making is particularly complex and controversial among individuals with serious mental illness (SMI) ([71]). The complexity is related to how psychological, social, cultural, legal, and ethical issues are deeply intertwined in a psychiatric system ([61]). The definition and threshold for diagnosing psychiatric illness tend to lack universal or clear objective definition, contributing to contention regarding capacity and need for SDM ([81]). Individuals with SMI often have fluctuations in their mental health stability and by extension, their capacity for treatment decisions, and the time needed for SDMs ([37]). Furthermore, some individuals with SMI may strongly disagree with a psychiatric diagnosis, and the views of others, and this conflict or disagreement affects their overall judgment towards treatment and need for SDM, contributing to a much-disputed process and impaired relationships ([27]). In addition, due to the often-protracted nature of mental illness, it may also be unclear whether a patient has had a prior expressed wish, or if that wish was made during a period of full capacity ([21]; [19]).

The substitute decision-making process in mental health further differs by jurisdiction. In Ontario, Canada for example, it is guided by three legal frameworks: the Mental Health Act (MHA), the Health Care Consent Act (HCCA), and the Substitute Decisions Act (SDA) ([44]). In short, the MHA regulates civil commitment, detention, and treatment during mental health crises. MHA deals with the criteria under which a person can be involuntarily detained in a psychiatric facility if that person is suffering from a mental disorder that is likely to result in serious bodily harm to him or herself or others, or serious physical impairment (e.g., inability to care for his or her basic needs). Involuntary detention does not automatically authorize treatment. A separate consent is needed for treatment under the HCCA. The health practitioner proposing the treatment—often a psychiatrist or physician—must determine if the person understands the information relevant to the treatment decision, and appreciates the consequences of making or not making a decision. Capacity assessments are decision specific and time specific. If a patient is found capable, he or she can refuse treatment, even if he or she is involuntarily hospitalized. If found incapable, the SDA and the HCCA authorize and compel a physician to seek out or appoint an SDM to deal with consent to treatment on his or her behalf. Of note, SDMs in Ontario can be appointed in two main ways—either by advance appointment by the person through a Power of Attorney for Personal Care (POAPC) so the person can choose someone he or she trusts and therefore feel less unfamiliar or coerced with the decisions, or, if no POAPC exists, the HCCA sets out an automatic hierarchy, so a spouse or partner, a child or parent, a sibling, or other relatives—in that order, whoever is available and willing first—can be appointed as SDM. A Public Guardian and Trustee (PGT) is used if no one on that list is available or willing. If an SDM resigns, the next person in the hierarchy who is available and willing becomes the SDM.

In terms of recourse, if someone is found incapable and disagrees, he or she can appeal to the Consent and Capacity Board (CCB). Under the HCCA, the SDM must follow the incapable person’s prior known wishes (when capable and applicable to the situation); if no prior wishes exist, they must make decisions in the person’s best interests considering that person’s well-being, known wishes, and weigh potential benefits vs. risks of treatment. SDMs cannot override clear prior capable wishes, and their decisions can be challenged (e.g., by the healthcare team) if they act improperly, and the case can go to the CCB. If SDMs are perceived to be not making appropriate decisions, clinicians can seek to challenge their decisions or replace the SDM with the CCB, although this is very rare. Relatedly, Ontario does not have a separate statute for psychiatric advance directives (PADs), but people can create advance directives through advance care wishes documented in a Power of Attorney for Personal Care, or even in writing/recording (informally) as long as it is clear and made when capable, and SDMs are bound to follow these prior capable wishes if applicable ([44]).

Like other fields of medicine, the SDM process in psychiatric care is created to protect vulnerable individuals, but it also creates special challenges for clinicians who must navigate the complex ethical, legal, and practical implications of decision making through a third party. As healthcare professionals strive to balance their duty to care for their patients, and the legal and ethical principles of beneficence, non-maleficence, and respect for autonomy ([4]; [32]), they often encounter unique situations and dilemmas related to the substituted judgment standard, the best-interests principle, and working with SDMs who have varied levels of involvement, style, and approach in care, as well as knowledge levels ([38]).

At the core, the key challenge clinicians face in this context is the tension between respecting the patient’s autonomy and working with SDMs to make optimal healthcare choices. The legalistic nature of substituted decision making could mean that the patients’ direct voice may be lost from the decision-making process, raising concerns about whether their wishes are truly honored ([3]). This can be particularly problematic if and when the SDM does not consult or reflect the patient’s views or wishes, or when there is ambiguity about the individual’s previously expressed preferences, or when the SDM’s decisions appear to contradict the known values of the patient. Clinicians must, therefore, engage in careful communication and collaboration with SDMs to ensure that the patient’s rights, dignity, and prior wishes remain at the center of appropriate clinical care.

Furthermore, the appointment of an SDM can introduce additional emotional and relational complexities for clinicians. Ethical conflicts often arise when family members or legally appointed guardians have differing opinions on the best course of treatment. When SDMs’ decisions are inconsistent with clinicians’ perceived best course of action for the patient, and disputes can occur over choice of treatment, or whether to use coercive and aggressive interventions vs. withdrawal of care—these conflicts invariably place clinicians in difficult positions. Working with both the patients and the SDMs, clinicians must mediate between the interests of the patient and the perspectives of the SDM ([8]). These conflicts can also contribute to moral distress among healthcare providers, affecting their professional well-being and clinical care.

Another crucial dimension to consider is the international human rights perspective, particularly the United Nations Convention on the Rights of Persons with Disabilities (CRPD), which advocates for the abolition of substitute decision making in favor of supported decision making ([74]). The CRPD challenges traditional guardianship and substitute decision-making frameworks by emphasizing the importance of providing adequate support to enable individuals to exercise their legal capacity. This global movement may have implications for clinicians, who to date are required to conduct clinical care using a capacity-based approach according to domestic law, but who may feel some moral distress in using coercion in their day-to-day work and are aware of the larger global call by the CRPD and some consumer activists to prioritize patient autonomy ([28]).

Despite the CRPD’s call for reform, the practical implementation of supported decision making remains a significant challenge in clinical settings, and remains unimplemented to date around the world. In many countries, such as Australia, Netherlands, Norway, France, and Canada, governments who have signed the CRPD have also signed a reservation to continue the use of substitute decision making ([20]). This reservation seeks to allow the current norm of practice to continue, as most clinicians contend that while some individuals may benefit from decision-making aids, communication tools, or advocacy support, others may have severe cognitive impairments that limit their ability to participate meaningfully in their care decisions ([24]). Clinicians must, however, be cognizant of this tension between supported and substitute decision making and develop expertise in assessing and enhancing decision-making capacity while continue to work closely with SDMs ([54]).

From many perspectives, mental health clinicians working with SDMs and patients who have SDMs face challenges at personal, clinical, and system/societal levels. Yet, the experiences of clinicians doing such work are little studied. There is also a paucity of comprehensive guidelines or education to equip healthcare providers with the skills necessary to navigate such a complex area of work—most learn on the job, and their experiences could be varied and worthy of a closer examination.

This qualitative study seeks to explore these important topics by examining the unique experiences of mental health clinicians who work with patients requiring SDMs, highlighting the ethical dilemmas, emotional burden, and practical challenges they encounter. By drawing on empirical research and theoretical frameworks, we examine how clinicians interpret their roles, and how they navigate the shifting landscape of decision-making authority. Ultimately, understanding these experiences can inform clinical strategies and potentially policy reforms that promote patient-centered care, balancing respect for the autonomy and dignity of individuals who lack decision-making capacity, and the need to protect and care for patients, in a time of potential global paradigm changes in this regard.

## 2. Methods

### 2.1. Study Overview

The primary research question guiding this study was: *What are the experiences and perspectives of clinicians working with SDMs and psychiatric patients who have SDMs?* We used a focus-group approach involving clinicians from both inpatient and outpatient psychiatry settings. To guide the focus-group sessions, we carried out a literature review to inform the development of a set of semi-structured questions to explore the clinicians’ experiences and reflections related to our research question. The study was generally informed by constructivist grounded theory. Constructivist grounded theory is a contemporary revision of grounded theory—a systematic qualitative research methodology that emphasizes the generation of theory rooted in data ([29])—that assumes a relativist approach, acknowledges multiple perspectives and realities in researchers and research participants, and takes a reflexive stance towards actions, situations, and outcomes of interest, also allowing the researchers to form hypotheses to inform the qualitative interviewing and data analyses and discussions ([11]). A focus-group approach was chosen as it provides a rich and in-depth exploration of an aspect of clinical experiences about which the interviewees have substantial subjective and shared experiences, and the group dynamics and synergy from the interactions in a group can promote exchanges, validation, and or the contrasting of ideas that can lead to novel insights and understanding. We also chose it because of the method’s flexibility, with real-time exploration and feedback of emerging opinions, experiences, and themes, as well as cost effectiveness ([41]).

### 2.2. Participants

Participants were clinicians who have substantial experience working with SDMs and patients who have SDMs, recruited from the inpatient and outpatient services at St Michael’s Hospital in Toronto, Canada. The hospital is a large teaching hospital in downtown Toronto, serving diverse social–economic and ethno-racial populations. Potential participants were screened by a Research Coordinator for eligibility and then invited for the focus groups. The inclusion criteria were (1) clinicians who have worked for three years or more with patients and SDMs; (2) over 18 years of age; (3) speaks fluent English, and (4) able to provide consent. There were no firm exclusion criteria. To maintain confidentiality, we have aggregated the social-demographic details of the participants as follows: Job roles included case managers (of various background, including social worker, occupational therapist, psychology majors, etc.), registered nurses, clinical team leads, psychiatrists, and care and transition facilitators (social worker background). The inpatients group had 5 participants, and the outpatient group had 6 participants. The ages of participants were mainly between 30 and 40, with a range from 25 to 55; and gender was about one-third men and two-thirds women.

### 2.3. Data Collection

We conducted two focus groups, one for inpatient clinicians and one for outpatient clinicians, each lasting about 60–70 min, in late 2019 to early 2020. Focus groups aimed for an in-depth narrative and discussions on understanding the experiences, reflections, and reactions to working with SDMs and patients who have SDMs. In addition, the focus-group questions explored the clinicians’ observation on the impact of the SDM process on family relationships, emotional reactions, decision-making process, and clinical outcomes. Based on our specific study question, we did not anticipate nor aim to explore any major difference between the in- and outpatient clinicians. The focus-group questions aimed at higher-level experiences and reflections; some sample questions for both the focus groups were *What is it like working with a client who does not have the capacity to consent to their psychiatric treatment? Tell us about what your relationships with your clients who require an SDM are like. What it is like to work with clients whose capacity in terms of their mental health fluctuate? What role do you find yourself playing? What are your relationships with family members who act as SDMs like? How do you manage disagreements about treatment with SDMs? What do you think is important for SDMs to consider when making psychiatric treatment decisions for their family members? What implications have you seen of having family members act as an SDM? What might be beneficial for clients? The care team? Client recovery? What might be challenging? What have you seen happen to the relationships between family SDMs and clients during an episode of care? How? What is it like to see a client transition from having an SDM to making their own psychiatric treatment decisions? What would you change about the SDM process if you could?*

As informed by qualitative research methodology ([49]), the probing questions were revised as needed during the focus group in light of emergent themes. The focus-group session process was flexible, and questions pursued important ideas, opinions, and emotions further during the session, specifically by asking “what” and “how” questions and eliciting examples ([11]). All qualitative interviews were taped and transcribed verbatim. The accuracy of the transcripts was verified by the research team.

### 2.4. Data Analysis

Our analyses examined how clinician participants experienced their role in working with SDMs and patients who have SDMs, constructed meaning and reflected in relation to these experiences through the line-by-line coding of participant narratives, and by comparing statements, emotions, and responses within and between participants ([11]). Coding was an ongoing process initiated during the data collection period to inform the continued development of “sensitizing concepts” from the data. All data collected from the sessions were coded using an iterative process wherein each focus-group transcript was read, coded, re-read, and re-coded as necessary by the first author, and then reviewed with the senior author. Analytic memos were also written iteratively after open coding to capture the major issues relevant to each code. In summary, this process occurred through three stages: (1) open coding that examined, conceptualized, and categorized the data; (2) axial coding that regrouped the data based on interrelationships within and among the original identified categories; and (3) selective coding that identified the ‘core’ phenomenon and major themes ([67]). Final thematic analyses were carried out by examining the analytic memos iteratively to capture and decide the final major issues most relevant to the research question; and these were then discussed together by the research team, with any outstanding issues resolved through discussions ([12]).

### 2.5. Ethical Considerations

This study was carried out in accordance with the Declaration of Helsinki for human subject research, and approved by the St. Michael’s Hospital, Unity Health Toronto Research and Ethics Board.

## 3. Results

A thematic analysis of the data showed the following themes and subthemes:

### 3.1. Clinicians Appreciate Hardships and Challenges in Lives of SDMs and Patients

#### 3.1.1. Clinicians Appreciating the Challenging Emotions and Experiences on Both Sides

Clinicians hold the role as a care provider through a biopsychosocial perspective, but also have the privilege of knowing and understanding the perspectives of those from both the patients and the SDMs. While they are treatment outcome oriented, they also feel the challenges and emotions of their patients and the SDM. About patient’s loss of autonomy, one participant (IP5) reported:

“*And I think for patients it can be difficult because for the patients sometimes it’s humiliating…, we were talking with a patient about making his uncle his SDM and the patient said I’m 40 years old like can’t I do this…. I felt badly for him as he felt humiliated that he wasn’t being treated as an adult but at the same time he’s not capable of making his own decisions at this point.*”

On the SDM side, clinicians note SDMs struggle with the emotional burden of overriding their loved one’s autonomy. One participant (IP3) talked about the challenges of being a family SDM:

“*…one of the stresses for families in being involved in making treatment decisions is appreciating the limits of what can be done and the protections available to people who are so ill but are accorded all these rights to determine their own healthcare when they’re ill. And it’s sobering for them to realize that they’ve lost control of trying to help their family member…, sometimes we don’t appreciate how shocking it is to families.*”

One clinician (OP3) also felt:

“*Family can be a lot more invested trying to figure things out and at the same time also puts a tremendous pressure on [themselves], and it can make it very uncomfortable for them if they live with their loved one, they made a decision that he or she doesn’t like it makes things very complicated.*”

#### 3.1.2. Clinicians Note Risks and Relational Changes from Being an SDM

Beyond discomfort and challenges, clinicians are aware of deeper risks and changes in family relationships. One participant (OP2) observed:

“*I’ve had family members where they were assaulted as a result of that… I had a gentleman who didn’t want injections and the mother who was the SDM made a decision to ensure they got the injections. And they came but when they got home they punched their mom out because they were so frustrated that you put me through this, you were trying to kill me.*”

Another clinician (OP2) noted how one family was burnt out and quit as an SDM:

“*…he didn’t want this and like he couldn’t handle it any more…, [and they] move out of the city. I’ve heard more is that they don’t want to be that involved [anymore].*”

One participant (OP5) talked about relational changes in the family:

“*I think from what I’ve seen more often than not it impacts the relationship negatively because there is that huge power shift. Right, like the power that used to belong to the client is now belonging to somebody else and so if the pre-existing relationship was already rocky, it usually gets worse from what I’ve seen and if their pre-existing relationship was better than there’s more opportunity for like a more collaborative approach and sometimes it still goes bad…*”

Sometimes clinicians even advise family SDMs to take a break from the role out of concern, as one participant (IP4) reported:

“*…when the patient has gotten unwell, they’ve become quite abusive to the substitute decision maker, and it’s been very, very stressful and distressing for that person. And I’ve seen here lots of times where the doctor while explaining to the SDM will sort of give that person an out and say you know what, you’ve done this but if it’s detrimental to your relationship or your own safety…*”

### 3.2. Clinicians Have Understanding of Patient Situation and Respect for Their Autonomy and Wishes

#### 3.2.1. Clinicians as Promoter of Autonomy and Mindful of Prior Wishes

Despite the fact that patients have an SDM because they lack capacity for treatment decisions, clinicians pay much attention to their needs for autonomy as a basic guiding principle, especially in the context of SDMs making decisions on their behalf. One participant (OP2) observed the loss of autonomy in patients:

“*There’s highs and lows and different people respond differently to SDMs, and you often do see the side where people feel that they have a loss of power based on having an SDM.*”

From this sensitivity, clinicians often work towards restoring some balance, as one participant (OP4) opined:

“*We work in conjunction with the client…, that’s the optimal thing but I’m not sure if they really have a legal status. Well before you’re psychotic what you would or would not want but hopefully the SDM would respect those wishes or at least recognize that that’s the person’s wishes.*”

Another participant (OP3) noted their appreciation when SDMs take into account patient’s wishes:

“*I think in general family tends to sometimes start to go based on the best wishes in the past but it’s not all the time. I think in general with SDMs if they’re a family member or the PG and T they try to go with the best interest or what makes sense.*”

#### 3.2.2. Clinicians Are Aware of Clients’ Fluctuating Capacity

Patients having fluctuating capacity—for treatment and general functioning—complicates working relationships and assumptions for both healthcare providers and SDMs. Such awareness makes clinicians both cautious, and hopeful. One (OP3) participant observed:

“*We have this constant fluctuation you know; in any other field of medicine you have dementia, then you’ve got dementia. You’re not going to suddenly become capable the next time you make a decision* versus *in a psychiatric patient. You can be pretty psychotic or you’re using a ton of crystal meth and you’re not making any sense today but in three days you can have a rational discussion…*”

Such awareness also makes clinicians pay ongoing attention to fluctuating insight level, as one participant (OP4) said:

“*Specific like it’s situational in terms of someone who has insight…, because then you come to think well the [lack of insight] is clouding over what a person is thinking and thinking rationally. I guess that’s where that SDM comes in case the person is incapable [at that time].*”

#### 3.2.3. Clinicians Are Key in Facilitating Communication

Clinicians are multidisciplinary, involving psychiatrists, nurses, social workers and other allied professionals. Depending on the training background, they carry out some shared tasks and some more specialized tasks. Nurses, for example, have a set of roles and impacts that are related to their most front-line contacts. Being in the middle between patients and family SDMs allows them to help with their relationships and communications. As one participant (OP5) said:

“*It’s helpful to facilitate. Like I would teach my client more skills on how to advocate for themselves in front of their family and would do role play.*”

Another participant (OP1) summarized as follows:

“*I think having someone that knows both sides and both peoples’ perspectives can really sort of enlighten that conversation.*”

#### 3.2.4. Clinicians Are Involved in the Process of Transitioning from SDM to Self-Determination

The process by which patients transition from having an SDM to making their own treatment decisions is a clinical focus. Clinicians often welcome this, as one participant (OP1) described a successful transition:

“*I think that is a very quick process; right, they’re deemed capable like they’re capable… With treatment decision I find it’s, they go in, you know the doctor says yeah, they’re capable, they’re capable…, so it changes very quickly…*”

However, there are worries about the transition as well. One participant (OP2) described a relapse after such transition:

“*I often see…, where they deteriorate again because part of that is a disbelief that they needed these medications. They became well and then didn’t believe they needed that support anymore and kind of cancels that. So it really depends on … how they actually view the treatment they’re receiving and when… an SDM leaves the picture.*”

#### 3.2.5. The Right of Patients Being Kept Informed of the Decisions That Are Being Made Even When They Are Still Incapable of Making Decisions

In any process, not forgetting that the patient is at the center of care is important. One participant (IP3) emphasized:

“*…about treatment so you know the worst thing to do is to declare someone incapable and only deal with the SDM without considering that the patient can appreciate some information and…, the capacity will change and so at some point in the admission. They may be making their own decisions and so you should relate to them as if they’re at some point going to be able to takeover.*”

### 3.3. Clinicians Have a Special Working Relationship with Family SDMs

The patients who have SDMs are by circumstance and definition not in possession of a good understanding or insight of their illness, and typically avoid or decline treatment. Clinicians and families are more aligned in their goals to promote treatment to improve patient health outcome.

#### 3.3.1. Clinicians See Supporting SDM as Part of Work

To improve patient care, integrating and supporting patients’ network is very good practice. One participant (OP5) conceptualized in a unique way:

“*Sometimes like the SDM is like another client as well. I think validation is key for their unique circumstance. Also, like providing just resources such as support groups and for families and various counselling that can just help, like go a long way in helping them figure out how to make their situation work for them and providing psychoeducation and stuff.*”

One participant (OP1) described how to facilitate SDM and family work:

“*Those little nuances of their perspective that they might be overlooking because there’s a lot of emotion and history involved. So really creating like a safe space where everyone can kind of sit down and have like an open discussion where I guess like emotions aren’t overwhelming the interaction.*”

#### 3.3.2. Clinicians See the Harm from Delayed or Denied Treatment from Their Own and SDM Perspectives

One of the significant concerns among the healthcare workers is the potential harm caused by delaying or denying treatment due to excessive focus on autonomy. Some patients deteriorate substantially while awaiting legal processes for treatment approval, as one participant (IP5) reported:

“*Legally, the patients can contest any form past a Form 1…, they have the legal right to be appointed a lawyer and have a mini hearing in this room. That can be a delay of about a week.*”

During this time, patients may become more unstable, engage in self-harm, or become violent. Some family SDMs may resort to taking their loved ones to other jurisdictions where forced treatment is easier to access, but often at great personal and financial cost. One participant (IP1) recalled:

“*We had family members who took their relative back to Africa where they would just get treated immediately… Some families ask if they can drive their relative across the border to the U.S. for faster treatment.*”

#### 3.3.3. Clinicians as Educators and Collaborator with SDMs

Clinicians hold a special role as educator of patients and SDMs about the mental health system and treatment approaches. This involves providing the societal and official perspective, such as explaining the concept of autonomy and intents of the Mental Health Act, as one participant (OP3) pointed out:

“*You need to keep explaining to [SDMs] that the goal is not for the person to make the decision I agree with or you agree with or the right decision in quotation. The idea is [the patient] get well enough to make the right decision for themselves or they’re well enough that they understand the consequences of what they do; so they may do something that we 100% disagree with but it’s not really up to us.*”

#### 3.3.4. Clinicians See Importance of Maintaining Professional Boundaries

While it is important to promote therapeutic relationships, it is also important to maintain boundaries and neutrality. One participant (OP1) opined:

“*In fairness to both parties, right, because you don’t want to be the person that’s now choosing sides and making decisions. If you go to the client, the family may feel like you’re not there for them. If you choose the family that might destroy that rapport you have with the client so it’s a family conversation…. So we don’t destroy any of our bridges.*”

One participant (OP5) elaborated as well:

“*I think when there is disagreements it’s really helpful to clearly define what our role is and what our position is and what we can offer and cannot offer and just setting boundaries of like what’s appropriate and what’s not appropriate.*”

### 3.4. Clinicians May Find It Difficult at Times Working with SDMs

#### 3.4.1. Difficulty Working with Over-Involved or Uninterested Family SDMs

Clinicians understand in principle the value and role of SDMs as a decision maker for patients who are incapable. They prefer working with both in a collaborative and supportive approach. Sometimes they encounter SDMs who are less ideal, as one participant (IP3) reported:

“*The worst situation is when the SDM doesn’t want to know anything about the treatment and just says ‘whatever you say doctor.*”

Another participant (OP3) found some SDMS overwhelming:

“*It wouldn’t be fair to say that dealing with the families especially the ones which are more heavily involved… can be very challenging cause the family can get very demanding…. a nurse … said if that mother of that client keeps calling me anymore I’m going to quit my job… I mean you want to allow the family involvement, to have family-centered care and client-centered care but …that showed how hard it was for her to deal with that.*”

In an extreme case, one participant (OP4) reported senses of vicarious trauma from working with patients and SDMs:

“*…you know this term vicarious trauma people have… And I guess well this reality is that the families, the emotions are so high, the love is so mixed up with all these other emotions and feelings …, it can be purely have an impact on you know, how you do your job and to the family members and it goes on…*”

#### 3.4.2. Clinicians Cope with Perceived Poor Decision Making by SDM

Clinicians often deal with SDMs’ treatment decisions they do not like or disagree, but they tend to work through with more education and patience, but feeling powerless to do much more. One participant (IP2) gave a colorful example:

“*It’s a problem cause they’re not really acting [as SDM]…, when they disagree with everything that you’re saying. I’ll be the SDM but I only want them treated with homeopathic medications.*”

Another participant (IP3) highlighted how he or she worked as best as they can, but very rarely they will try to petition to change the SDMs (e.g., by demonstrating that SDMs are not acting in the best interest of the patients):

“*You can declare the substitute decision maker incapable but we rarely do that especially if they’re going to live with that family member because they’re going back into a system where the treatment is not going to be supported…*”

According to the current Ontario system, the SDMs are supposed to base their decisions on the patient’s prior expressed wishes. However, in some instances, the clinicians feel that the patient’s prior wishes are not being respected, as one participant (OP3) recalled a treatment-related decision by an SDM about resuscitation or not in the event of a catastrophic health crisis that went against the known prior wishes of a very frail elderly family member:

“*We had a situation recently where an SDM was providing consent for a very extreme treatment for their aunt who was very ill. So, specifically it was to provide full resuscitation in the event that this person’s heart stopped and he said I want you to do everything. Don’t pull the plug. And the ethicist said okay, but are you acting according to the prior wishes of your aunt and it was clear he wasn’t. This was his personal belief and so those are tricky.*”

#### 3.4.3. Clinicians Sometimes Ponder if SDMs Are Necessary

Given the variable involvements and challenges in working with SDMs, one clinician (IP5) wondered about the necessity of SDMs, while acknowledging and understanding the legal framework behind having SDMs:

“*But I guess basically what I’m saying is like if there is no difference then what’s the difference between us just making all the decisions and not having even substitute decision makers… What is the benefit of even having these discussions at all…, is there actually utility to having an SDM or is it just sort of like a legal thing?*”

### 3.5. Clinicians Delineate Differences Between Family and Public Guardian and Trustee SDMs

#### 3.5.1. Clinicians See PGT as Closely Aligned with Medical Decision Makers

In this research jurisdiction, PGTs are a form of government-appointed SDM when next-of-kin SDMs are not available or not willing. There are strong qualitative differences between the two and in turn their interaction with clinicians. One participant (OP2) noted:

“*Well I think with a PGT it’s a pretty straight forward process from my experience; right. The doctors will make recommendations for different things… [PGTs] are usually in agreement so it’s pretty smooth transition* versus *the family where sometimes you know the family can have underlying mental health issues too so you know getting there sometimes it can be tricky but I think PGT definitely works a lot better and the process is a lot quicker as well.*”

Another participant (OP1) added:

“*… with the PGT I think you can find that it’s almost as if the treating team is the substitute decision maker at that point because they are so amenable to sort of listening to the healthcare professionals and sort of going along with that. Because I don’t know if they have the same sort of expertise of buy-in. Whereas a family they might look beyond just exactly what the healthcare professionals are proposing which I’m not saying one is better than the other. I’m saying that it really changes the way care is driven.*”

#### 3.5.2. Clinicians See Family SDM Intimately More Involved

Clinicians observed that family SDMs operate very differently from PGT SDMs. One (OP5) observed:

“*With family I think you just feel a lot more of the emotion involved in it and in making decisions.*”

Another participant (OP2) also noted:

“*I also just think of the aspect of having a family member allows an individual to probably maintain their personhood more consistently. I think a lot of the times our mandates and the things that we’re trying to accomplish might overlook how a person might actually live and function based on what we think is a state of wellness, where we might not actually see them as a person…, where a family member might be able to say like this isn’t really this person anymore.*”

#### 3.5.3. Family Is More Likely to Disagree with a Physician’s Recommendation in Cases Where They May Perceive the Treatment as More Extreme

More on the difference between family and PGT on perceived invasiveness, as one participant (IP3) described:

“*I think I’ve practiced like 24 years. I’ve never had a PGT say no, I’m not consenting that. I’ve had PGTs consult lawyers and see people regarding really coercive treatments like ECT but I’ve never had a PGT say no. I’ve had lots of families say no.*”

### 3.6. Clinicians Recognize the Importance of SDM Role in Many Contexts

#### 3.6.1. Clinicians See Social Values in Having SDMs

Psychiatry has been a branch of medicine that has many controversial aspects in public discourses, historical and present. This backdrop informs clinicians’ views and attitude towards SDMs. One clinician (OP4) observed:

“*I think the family, the SDM component as well is definitely like a reflection of the past where you know without that SDM, without someone who actually cared for them these individuals could and often were sort of kept detained and treated you know and not always in the most equitable way. Right, so it’s having those things in place allows society to say okay, like there are stop gaps to prevent harm from being done. Unfortunately, we’ve had those historical sort of tragedies happen and these things have been put in place as a result.*”

Another participant (IP3) articulated:

“*Well society is really split. They are very ambivalent about psychiatric treatment. They don’t want homeless people on the street. They don’t want psychotic people killing people which happens and so they say do something about it. At the same time, they want to limit the powers of that and health teams come in because there’s a history of bad treatment, taking it too far….*”

#### 3.6.2. Clinicians Feel Having SDMs Help Them to Feel Better About Their Actions

Treating patients who are incapable often involves emotionally draining work, as treatment is not wanted or understood by patients, and sometimes leverages and formal coercion using Mental Health Act-based forms and mandates are involved. SDMs who support these approaches are helpful as they make clinicians feel less guilty or morally distressed. As one participant (IP3) reported:

“*Yeah, if we can see the harm that’s ahead just because it’s uncomfortable to have to have those adversarial conversations. I will be forcing this because someone else has consented. It’s the goal and the prevention of future harm.*”

#### 3.6.3. Clinicians See SDMs as Protective for Patients

Overall, clinicians feel SDMs are helpful and address a gap in the system, as one participant (OP1) observed:

“*I think the SDM component is definitely like a reflection of the past where you know without that SDM, without someone who actually cared for them these individuals could and often were sort of kept detained and treated you know and not always in the most equitable way.*”

### 3.7. Clinician Ideas on How to Improve the Current System

#### 3.7.1. Improving at a Public and Societal Level

With first-hand experiences, clinicians are well equipped to give feedback on how to improve the current mental health system and related issues with SDM. One participant (IP1) suggested social changes:

“*I think a lot of these… were based on the fact that there was a feeling that people were being hospitalized against their will and some sort of social control you know but at this point… we don’t want people… We are not holding people against their will that don’t have to be here. We don’t need to detain people for reasons that aren’t because they actually are a danger to themselves and others and…unable to make treatment decisions. It’s not a coercive…, the system isn’t abused the way maybe it was in the 60’s… So, I think that that process could get more modern.*”

#### 3.7.2. Improving at the Family SDM Level

Some suggested that family SDMs might have a better experience if they are more aware of the needs and experiences of their loved ones. There is room in supporting families and SDMs, as one participant (IP5) opined:

“*Families could know more or be updated better… The family members should know more about [their ill family member’s] wishes technically and know more about the situation.*”

Clinicians thought the mental health system and clinics could do more to educate families about what SDM role is about, as one participant (OP3) suggested:

“*I think part of the problem is also there isn’t really a good process that explains to people what their rights are as an SDM, what is it that they can ask for, what are the things they’re entitled to. It’s just kind of pushed upon somebody to act in that role and then they can be frustrated. Why aren’t you Forming my son and bring him in—because I can’t, there’s no reason, there’s no ground to do that.*”

At times family may feel that because they have been given the title of substitute decision maker, they now can control many aspects of the patient’s care, not realizing that their role is limited to only making decisions with respect to patient’s psychiatric care; as one participant (OP3) pointed out:

“*I think sometimes there’s a lot of un-clarity on their part even though they’re explained. They may be the SDM for treatment but not necessarily for finance and not necessarily for something else.*”

Another clinician (OP6) echoed:

“*I think it would be beneficial for more formal education for the SDM to sort of outline their role…. Say you know what just so you know you’re agreeing to be an SDM so we have to read you this thing just so you know your rights and responsibilities and at the end you acknowledge you know you accept it or not which is like mandatory.*”

## 4. Discussion

The current study of clinicians working with patients and their SDMs contribute to this little-studied field. The situation where a patient lacks treatment decision capacity and requires the involvement of an SDM is unique and carries clinical, emotional, medical–legal, and societal significances. The clinicians’ experiences and perspectives as summarized in the thematic analyses show they appreciate the hardships and challenges in the lives of SDMs and patients, understand the compromised situation of the patient, and have respect for their need for autonomy. In addition, they want to have their wishes fulfilled, have a special working relationship with family SDMs, but at times find it difficult working with them. Moreover, clinicians are in the best position to appreciate and delineate the differences between family and Public Guardian and Trustee SDMs, recognize the importance of the SDM role in various contexts that only front-line workers would, and contribute ideas on how to improve the current system.

Working with both patients and their SDMs is a unique position, and gives clinicians an intimate proximity to appreciate the challenges and difficulties involved in being the patient and the SDM. There are many emotional and cognitive reactions to being in this position. Clinicians’ perception that SDMs perform a very hard role is well founded. Research shows that SDMs for patients with critical medical illness ([75]), or severe cognitive or mental illness alike can result in substantial emotional burden ([43]; [21]; [66]). The SDM role can be so challenging that one medical SDM study showed many SDMs experience post-traumatic stress reaction, consistent with a moderate to major risk of Post-Traumatic Distress Disorder (PTSD) ([2]). Research on SDMs for people with mental illness often reported more frequent and serious family disputes and high emotional burden ([63]). In addition, mental health SDMs often feel there is inadequate support from healthcare providers and the system at large ([25]). However, they see their role as highly valuable, and their commitment is high ([79]; [56]; [76]).

As seen in this study, clinicians appreciate that family SDMs are an integral part of the current medical–legal system in psychiatric care ([16]), and what they witness as a caretaker burden on SDMs is unfortunately all too commonly found in other studies as well ([77]; [59]). Clinicians are aware that prolonged involvement as an SDM in caregiving can lead to burnout, and therefore clinicians are very understanding that some SDMs may eventually step back or refuse to continue.

Clinicians also appreciate the emotional and psychological impacts that having SDMs can have on patients, as their autonomy is fundamentally affected. Research shows that a minority of patients may welcome SDM involvement ([47]), but many factors contribute to patient dissatisfaction and conflict with SDMs and clinicians, including a poor definition and threshold for diagnosing mental illness and assessing capacity ([81]; [27]), fluctuations in mental health stability, level of insight into one’s illness (from a medical professional’s perspective) ([37]), and disputes between one’s prior expressed wish made during a period of full capacity versus SDM-judged current best interests, among others ([21]; [19]). As the clinicians in the current study found, disenfranchised patients contribute to the challenges encountered in serving patients who have SDMs. Ultimately, these difficulties in patients and SDMs tend to lead to heightened frustration and resistance in all parties, making it harder for healthcare providers to maintain therapeutic relationships with patients.

Relatedly, clinicians are keenly aware of the importance of respecting patients’ autonomy and prior wishes, especially when their autonomy has been affected by the existence of SDMs ([47]). In the context of such compromise, clinicians are mindful of finding ways to promote as much autonomy and respect as possible. In that process, clinicians provide an ethical and sensible safeguard in ensuring that SDMs are making decisions in the patients’ best interest, and honoring patients’ wishes, especially if those wishes were expressed prior to their incapacity. Nevertheless, mental health professionals must sometimes override patient autonomy by offering and implementing treatment to prevent long-term harm. It is important to recognize that this is a highly controversial aspect of psychiatric care. As illustrated in the current study, some practitioners may feel forced treatment is paternalistic and infringing on patient rights, while others believe that failing to treat an individual in a timely fashion due to short-term resistance can lead to unnecessary suffering and long-term deleterious effects ([68]).

In the day-to day-work in our jurisdiction, the current study highlights the situation where the clinicians are at times caught in between patients and their SDMs in a system that is quite adversarial. The legal system confers the psychiatric treatment decisional power in theory in absolute terms, either with the patient, if capable, or with the SDM, if incapable. Sometimes patients may have partial insight into their illness but do not recognize the need for an SDM. This in turn can create friction between the patient, his or her SDM, and the healthcare team. Healthcare providers encounter these often and experience this tension. To address this adversarial status quo, many have advocated supported decision making as promoted by the UN CRPD ([14]). Supported decision making may involve advanced treatment directives, joint crisis planning, timely counseling, expert support, etc., that can potentially decrease the level of coercion in treatment, improve clinical outcome, and make clinicians’ jobs easier overall ([6]; [35]; [17]). One such concrete example is Open Dialogue, a Finnish model that prioritizes the individual’s ability to make their own choices, while providing support to ensure informed and voluntary decisions. Open Dialogue involves network meetings where mental health professionals facilitate the dialogue, involving the patient, their chosen network, and healthcare professionals to ensure everyone’s perspective is heard and valued, focusing on transparency and positive regard, aiming to collaboratively address treatment and other recovery related issues, emphasizing support in people’s homes and communities to avoid hospitalization ([5]).

However, none of these approaches are perfect, especially in the context of SMI ([54]), as patients may often not be able to decide well enough even with ample support. The CRPD approach on abolishing substitute decision making altogether also steers this to the opposite absolute, with only patients ultimately deciding on all treatment decisions and relegating family and others to a supporting role ([13]; [60]). Families and clinicians alike would naturally be wary of this more purely rights-based approach, as they fear the patients who are incapable may make the wrong decision such as refusing treatment at critical moments in their illness due to the chronic effects of the illness itself, or a lack of insight, or the acute effects of intoxication, psychosis, or dementia ([61]). In reality, something less absolute, where a more flexible decision-making process based on both rights and capacity may be a solution that should be explored ([64]).

Having experienced the tension and challenges in working with SDMs, clinicians were the best evaluator of the two different kinds of SDMs currently available in our jurisdiction. Clinicians observe that family SDMs have greater understanding of the patients’ history, needs, and personal habits, with strong emotional investment and ties to the clients, more likely to be worried about potential harm, and more attentive to patients’ desires. Therefore, clinicians feel family SDMs could possibly provide more consistent and continuous care that potentially allows patients to feel more supported and understood. For a minority of family SDMs, there had been concerns over their availability and reliability, highlighting the fact that an SDM’s role could be less meaningful if he or she is not actively engaged. Most research shows that family SDMs are highly dedicated ([46]), if not too involved due to a “role captivity” phenomenon as both a caretaker and the SDM ([70]; [45]). On the other hand, clinicians see PGT SDMs as more impersonal and procedural, which affects the ability of patients to develop a bond with their SDM. However, this may also allow PGT SDMs to make difficult treatment decisions in an impartial manner where family SDMs may falter. Having family members as SDMs could help patients maintain their dignity and identity in ways that professional SDMs may not be able to. The clinicians’ observations highlight a significant limitation of the PGT system in maintaining personal and responsive relationships with patients, and these views are valuable and informative, as no known study has contrasted these roles.

The study also finds that clinicians sometimes do not work easily with family-member SDMs. Other than emotional over-involvement ([7]), one key such reason is that clinicians sometimes perceive family SDMs as not well informed about treatment options, or may not act in patient’s best interest as perceived by clinicians, creating conflict and friction between SDMs and clinicians. When this occurs, and if the usual psychoeducation did not help to change the SDMs’ minds, clinicians often acquiesce to the SDMs’ decisions, as it is impractical or too legally cumbersome to challenge them. These kinds of ethical dilemmas encountered by clinicians are stressful, and the decision not to challenge SDMs is often a difficult choice (only in extreme cases would the healthcare team challenge the SDM’s capacity). While this is a little-studied area in psychiatry, the dilemmas echo other fields of medicine that often deal with such disagreements more acutely, for examples, in intensive care units, or palliative care decisions ([85]; [42]). In these situations, research in other medical settings has shown that closer working relationships, giving counsel, and finding common ground and collaboration are best solutions ([73]; [36]). More training and education for clinicians on how to work with SDMs may also be needed. The use of a mediator or intermediary counselor may also be potentially useful. Without such training or resources, clinicians may feel that they cannot effect change and end up feeling powerless and ineffective, and sometimes they even lament and wonder if the system—and in turn the patients—would be better off without the SDM. These are genuine reflections that highlight the difficulty in carrying out patient care through an SDM.

In addition, clinicians find the role confusion in family SDMs challenging. Clinicians recognize that family SDMs may feel overwhelmed, and often feel the need to assert their views through frequent contact with healthcare providers. Many family SDMs tend to assume they have broader authority than they actually do, and some may mistakenly believe they can decide non-medical aspects of the patient’s life as well, as a parent would typically do for a child. As a result, healthcare teams often struggle to navigate family SDM expectations, especially when they try to influence treatment beyond their legal scope. These misunderstandings could lead to conflict with healthcare teams. This finding echoes reports from research on family SDMs’ own experiences, where they feel they have too limited “authority” to care and decide for their loved ones ([46]). More education and support for family SDMs seems to be strongly advocated by front-line staff.

Unlike conditions such as dementia and intellectual disability, a central issue in psychiatric care is that decision-making capacity is less static. This fluctuation in capacity makes it difficult to determine how long the current state of a patient will last, when an incapacity decision should be revisited, and when and how an SDM should be involved ([37]). Clinicians have to exercise expertise and judgment for these decisions, while managing the pressures from patients and family SDMs who would like to have their points and wishes come to the forefront. This shifting nature of work is part of the challenges and difficulties in working with the SDM system. A good practice is to have regular and longitudinal capacity assessments built in, providing support and psychoeducation for patients to make the best decision for themselves, and respecting the decisions patients make when they are capable ([52]). When a patient becomes capable of making treatment decisions, it is still something that causes both celebration and worry for some clinicians ([39]). While it is the nature of the work, as many clinicians have acknowledged, clinicians could also learn to accept the outcome of patient’s autonomous decisions more ([81]; [27]).

Clinicians experience and witness the tension and potential harm that can arise when patients disagree with their SDM’s decisions. These conflicts can lead to physical or emotional harm, creating risks for both the patient and the SDM. In addition, patients who resent having an SDM often redirect their frustration onto clinicians, viewing them as enforcers rather than caregivers. This can create hostile interactions, making it difficult for clinicians to maintain trust and rapport; thus, there is potential direct and vicarious psychological trauma or harm for healthcare workers ([23]). The concept of “vicarious trauma” describes how those who care for individuals with mental illness may internalize the emotional weight of the situation, potentially influencing their own mental health and their ability to support the patient effectively. Clinicians in the current study are mindful of these potential risks in their interactions and reflections, suggestive of possible burnout, if not full-scale trauma ([18]).

Despite the various challenges, clinicians also see many positive roles the SDMs perform, and they report and feel they have a unique working relationship with family SDMs as their views and goals may be more aligned due to a shared hope for the potential benefits of treatment. Clinicians sympathize with the family and bemoan delayed or denied treatment; they carry out extra psychoeducation for SDMs, and they see families often as informal allies ([34]). There is strong evidence that timely treatment for severe mental illness, for example, schizophrenia, would both decrease the immediate morbidity associated with the illness, and prevent detrimental changes related to prolonged untreated psychosis. Treatment-related improvement would have positive impact on overall psychopathology, level of functioning, and cognitive functioning ([82]; [84]). Nevertheless, clinicians also frequently struggle with ethical concerns about imposing treatment, often coercively through Mental Health Act-related mandates (e.g., Community Treatment Orders or Outpatient Commitments, or involuntary hospitalization orders) ([55]). As this study shows, clinicians help family SDMs work through many challenges in managing complex choices as counselors, educators, and partners ([40]). Research shows that SDMs often feel that they are not being included or integrated in patient care enough ([69]). Acknowledging, respecting, and involving SDMs seem essential. In this mutual relationship, clinicians also report being comforted by the fact that having the support of family SDMs gives them some reprieve in the moral distress their work may incur in their relationship to coercive treatment approaches ([22]). Of note, the concept of procedural justice provides very useful principles to guide such complex interactions in the special role that clinicians play. Procedural justice advocates transparency, respect, empowerment, and fairness in dealing with vulnerable populations—relevant, as clinicians in this study are working with those who have been deprived of their treatment autonomy ([26]; [51]).

The transition process from having an SDM to self-determination is an important step in patients’ recovery, and clinicians reported both positive welcoming attitudes, while holding some worries of relapse if the patients neglect treatment after regaining capacity. These mixed feelings are common and reality-based in working with people with SMI. From the framework of a recovery journey, clinicians’ wish to use a risk-avoidant approach has to be balanced by realizing the importance of regaining autonomy, dignity and the lowering of stigma that stems from having an SDM or perceived coercion from the patients’ perspective ([65]; [48]). Clinicians could also benefit from understanding the notion of “dignity of risk”, where patients see positive quality of life gains even if they take on risk in their choices ([10]).

Clinicians are very mindful of the impact of public perception on the treatment of people with SMI. Several clinicians discussed that society’s historical mistrust—related to past abuses in psychiatric institutions—of psychiatric care further complicates the conversation about involuntary treatment and working with SDMs for them ([33]). Clinicians report that they are cognizant of the paradox in public attitudes—society expects mental healthcare providers to prevent crises but is uncomfortable with the legal authority needed to enforce treatment. This tension is also embodied in the current political debate about substitute decision making. While SDMs are the norm in current Canadian practice ([9]), many Canadian advocacy and legal organizations, such as the Law Commission of Ontario ([3]), the Council of Canadians with Disabilities ([15]), and the Canadian Association of Community Living ([15]) all supported the removal of SDMs, making the task of continued use and working with SDMs more contentious. As the study shows, clinicians are acutely aware of the historical backdrop that resulted in the current system, and this knowledge seems to be as essential as clinical expertise for carrying out psychiatric care in a balanced, aware, and still compassionate way ([72]). Of course, beyond the historical reason the clinicians raised, there are restrictions on clinical interventions simply because they represent a significant incursion into a patient’s civil rights and are in constant tension with the social and human rights values of liberty and bodily integrity as a matter of principle. Involuntary detention and psychiatric treatment and substitute decision making have always been controversial, and there exist many different and valid perspectives and interests within the community ([78]).

Clinicians working on the front line also have unique perspectives on how to improve the current system involving SDMs. There was agreement around the need for better education for SDMs, especially to ensure that they understand the extent and limitations of the role, and responsibilities. The current system also seems to lack sufficient information and support for SDMs, which can lead to confusion and difficulties in fulfilling their duties, and clinicians having to fill this gap in their interactions with SDMs and patients, creating unnecessary conflict and tension in the process. Clinicians call for formal education and clarity around the responsibilities of SDMs, and support for performing this important role. These suggestions echo those perspectives coming from the family SDMs themselves ([46]), and research at large ([62]). Furthermore, reconciling the larger clinical practice changes advocated by the CRPD is an important transformation that must involve clinicians who are an integral part of any such reform ([58]; [57]).

Based on this research, the current study’s unique front-line experiences and reflections from clinicians working in the complex and nuanced intersection of patient care involving SDMs provide insight into the core issues of autonomy, duty to care and protect, advocacy, and the pros and cons of substitute vs. supported decision making. Our discussion also highlights the fact that clinicians are concerned that very ill individuals cannot be counted on to process information and may suffer irreversible deterioration if poor treatment decisions are made due to anosognosia; and on the other hand, patient consumers feel their autonomy is threatened by SDMs, and other important organizations in Canada have called for the end of SDM. Many more dialogues and research on these important topics are warranted.

The current study has a number of limitations. While we interviewed a representative sample of inpatient and outpatient clinicians in focus groups, they are from one hospital, and the generalizability of the study findings is limited by the jurisdictional and service locations that sourced our participants. Our sample size is modest and the number of groups limited; therefore, data completeness and saturation may be limited, and there may also be some selection bias, as those who came forward to participate may have more favorable or unfavorable experiences related to working with patients and their SDMs.

## 5. Conclusions

The current research reveals that clinicians have an important place in understanding the different roles and experiences patients and SDMs have. In addition, having a mediating role in many instances is a privilege that gives rise to opportunities to improve patient and SDM experiences, clinical outcomes, carry out education, and advocate for ethically just decisions. This inimitable clinician role also comes with frustration, discomfort, moral distress and at times vicarious trauma. Clinicians’ unique understanding of this particular, complex, and nuanced intersection of patient care involving SDMs gives insight into the core issues of autonomy, duty to care and protect, advocacy, and emotional dynamics involved in this important work. The results here can inform the clinical reality and possible improvement in this sector, as a larger philosophical and social movement is under way—a movement mandated by the UNCRPD that calls for the abolishment of substitute decision making.

## Data Availability

The data presented in this study are available on request from the corresponding author. The data are not publicly available due to privacy or ethical restrictions.

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
