# Peer review of "“Get Well Enough to Make the Right Decision for Themselves”—Experiences and Perspectives of Clinicians Working with People with Serious Mental Illness and Their Substitute Decision Makers"

_behavsci, 2025, doi:10.3390/bs15050704_

Round 1
Reviewer 1 Report
Comments and Suggestions for Authors
Thank you for inviting me to review this paper entitled “Get well enough to make the right decision for themselves” – Experiences and Perspectives of Clinicians Working with People with Serious Mental Illness and their Substitute Decision Makers. I enjoyed reading this well-written paper and I recommend you accept it for publication. However, the paper will be strengthened by the following changes/clarifications:
- I think that in the introduction the authors need to write a new paragraph defining what they mean by the terms ‘substituted decision-making’ and ‘supported decision-making.’ These things occur on a spectrum and I’m not sure that the authors understand the nuances here. At one end of the scale, substituted decision-making refers to decisions which are made on a person’s behalf by a third party in their objective medical best interests. At the other end of the scale, supported decision-making refers to situations where the decision is made by the patient after being offered informal and formal supports and if after substantial efforts have been made a person is unable to make a clear decision themselves then a decision will be made for them by a third party who will base it on the best interpretation of their will and preferences. That is, a substituted judgement model (see definitions in CRPD Committee’s General Comment 1). It seems like the model in Ontario is a bit of a hybrid of the two in that the substituted judgement approach is used (because the SDM is supposed to base their decision on the wishes of the patient and what they would have would have wanted, rather than on what the medical team thinks is in the patient’s best medical interests). There are also lots of possible variations along the scale from substituted decision-making to supported decision-making, as it is possible to have a substituted decision-making system which also uses supported decision-making to help patients who have partial mental capacity to be able to make an autonomous decision. Further, just because an SDM is appointed doesn’t mean that the SDM can’t talk to the patient even if they have lost mental capacity and take into account their views to the extent they are able to do so (in fact they probably should consult that patient where possible so that the patient understands what is happening and isn’t left out of decision-making). The SDM is still the legal decision-maker and gets the final say, but they can do so after taking on board the patient’s views and everything else they know about the patient. Therefore, ‘the patient’s direct voice’ doesn’t need to be ‘absent from decision-making’ (line 73-74). I think it might help if the authors do some further research. I suggest Kay Wilson, Mental Health Law: Abolish or Reform? (OUP, 2021) Chapters 1 &8 as a starting point.
- In the first paragraph the authors state that ‘The processes used to appoint and enact surrogate decision-making may differ across jurisdictions, but there are generally more similarities than differences.’ I have two problems with this statement. First, I think that the authors should be consistent with terminology and say ‘substituted decision-making’ rather than ‘surrogate decision-making.’ Second, the system of appointing family members and public guardians and trustees as SDMs in Ontario is quite different from substituted decision-making used in other jurisdictions. For instance, in the United Kingdom and Australia the SDMs are usually clinicians and/or mental health tribunals, so those clinicians can treat the patient more directly and do not have to deal with family members or public guardians. Therefore, many of the dilemmas in this paper simply do not occur for them. The Abstract also needs to be amended to make it clear that the process of substituted decision-making in Ontario is not used everywhere.
- In the introduction I also think it could be useful to clarify that the paper is about psychiatric patients (rather than general medical patients who lack mental capacity) and briefly explain the legal framework for involuntary detention and psychiatric treatment in Ontario (presumably under the Mental Health Act) for international readers. I found myself wondering for instance (i) what the civil commitment criteria in Ontario are and whether there might be a difference between the power to detain someone in hospital and to treat them, (ii) how SDMs are appointed and by whom (for instance, can the patient appoint them by using a power of attorney which could be seen as being less coercive and closer to supported decision-making), (iii) how is mental capacity assessed and by whom?, (iv) what are the legal limits of SDMs?, (v) are there any processes for the making of psychiatric advance directives? and (vi) what happens when an SDM ‘quits’? If the authors are confused about this they could approach a Canadian mental health lawyer to become a co-author.
- I’m also not sure the authors really understand the CRPD and the role it plays. The authors state that ‘this global movement has significant implications for clinicians, who to date carry on clinical care using a capacity-based approach by societal and medical traditions, but must be aware to reconcile their day-to-day clinical responsibilities with emerging legal and ethical norms that prioritize patient autonomy.’ That isn’t quite correct. Clinicians don’t have to pay any regard to the CRPD when making their day-to-day clinical decisions. What they absolutely must do is obey the domestic law within their jurisdiction and their professional ethical duties. The CRPD is an international human rights treaty which isn’t directly enforceable in most domestic jurisdictions. It is instead the responsibility of states to implement the CRPD by incorporating it into domestic laws and policies. Now the interpretation of the CRPD and whether it actually requires states to abolish substituted decision-making is a very controversial question and as the authors have noted many states have interpretive declarations stating that they don’t interpret the CRPD as requiring the abolition of mental health law. In fact, no state has completely abolished mental health law to date. But, the CRPD does provide states with guidance about what laws and policies they should have and maybe one day some states will go so far as abolishing substituted decision-making (although I think it unlikely). The relevance of the CRPD for clinicians is perhaps more so in terms of working with government and consumers at a policy level to see if the relevant law reform can happen to reduce or abolish coercion in psychiatry, rather than something clinicians should be worrying about on a day-to-day basis. Clinicians might wish to try to pay more respect to autonomy, but they don’t have to – ultimately that decision is for parliament taking into account the CRPD and the views of a whole lot of stakeholders (eg. clinicians, families, consumers and the community in general). It is not for clinicians or those disability activists and those consumers who are advocating for abolition (which isn’t all of them) to decide.
- In the methods section describing participants perhaps you could clarify that the patients are psychiatric patients. Is it correct that there were only 11 participants in total? Why so few? What efforts were made to ensure that data reached saturation? Why was the data collected so long ago in 2019 and 2020? Did you use software to assist with the analysis?
- What are the problems with delaying treatment specifically? Does it prevent recovery?
- In reading about the way clinicians are balancing the interests of the patient and families I did wonder if the authors have heard of things like Open Dialogue and other models of supported decision-making that are thought to be non-coercive? See for instance Russell Rozinskis and Chloe Rourke ‘Challenging Involuntary Treatment and Confinement in Canada Through the United Nations Convention on the Rights of Persons with Disabilities (CRPD)’ (2024) 18(3) Studies in Social Justice 418.
- I’m a bit confused about the reference to non-resuscitation at lines 410-411. Does this happen with psychiatric patients?
- Were there any differences between the views of participants who worked in outpatient and inpatient settings?
- In the discussion the authors state that ‘When this occurs, the sense of discomfort is often dealt with through practical acquiescence.’ What does ‘practical acquiescence’ mean? ‘Acquiescence’ by whom? In this section I was wondering why there is reluctance to challenge decisions in that this would give everyone guidance as to what should happen and take the decision out of the hands of clinicians and SDMs at odds with each other? I also wonder whether there should be other means of dealing with these tensions such as the use of mediators or counsellors to work through conflict.
- The authors have some concerns about patient’s fluctuating capacity and how often capacity should be assessed. There’s also some discussion about dementia being more permanent. But, the answer is that capacity should be assessed regularly in relation to each specific decision (even for dementia patients) as we no longer take the view that patients lose global capacity once they have a particular diagnosis. Mental capacity is specific to a particular decision and a particular point in time. If a patient is assessed to have capacity then you need to respect their decision the same way you would for decisions made by a patient with a physical illness – if they relapse, they relapse. It is the role of the clinician to educate a competent patient and make sure they explain the relevant information and risks properly in a way the patient can understand. The clinician can also ensure they have access to supports. After that the ball is in the competent patient’s court.
- In the discussion on p 15, I’m not sure that the reason that there are restrictions on clinicians is simply because of historical abuses in psychiatric institutions. That is part of it, but the other reason is that coercion in psychiatry really is a significant incursion into a patient’s civil rights and is in tension with social and human rights values of liberty and bodily integrity as a matter of principle. Involuntary detention and psychiatric treatment has always been controversial and it still would be even if there were no abuses of power by psychiatrists and no scandals. It isn’t just about society’s mistrust, it is that there are simply a lot of different perspectives and interests within the community on a whole range of issues from whether mental ill-health is an illness at all or something that is a social problem and whether and to what extent society should enforce treatment and when. This is all really hard stuff with no easy answers and that’s okay. Perhaps some discussion of the CRPD could be useful here. See Kay Wilson, Mental Health Law: Abolish or Reform? (OUP, 2021) chapters 2 & 3.
- I did wonder whether one of the advantages of using family members as SMDs is that family members provide persons with mental illness with a lot of support for their long term recovery and it means that the family members are much more included in treatment decisions. In other jurisdictions, family members often complain that they are excluded from medical decision-making and that this can set the recovery of their loved one back.
- There are a few minor typos. Sometimes there’s confusion between the use of the singular and plural. In line 49, I think it would read better if it said ‘their own treatment decisions’ rather than ‘one’s own treatment decisions.’ At line 527 ‘the situation where a patients lacks treatment decision capacity’ should be ‘patient’ (singular). At line 739 it should read ‘who are an integral part.’
Reviewer 2 Report
Comments and Suggestions for Authors
I am troubled that this article focuses on "substitute decision making" (which I will call SUB) without enough attention on "supported decision making" (which I will call SUP). And I find this surprising because the authors do reference the Convention on the Rights of Persons with Disabilities (CRPD) (e.g., see Article 12). The thing lacking in this article is when and how the decision is made to use SUB instead of SUP.
In the very first example (line 212), there is a quote "...for the patients sometimes it's humiliating...we were talking with a patient about making his uncle his SDM and the patient said I'm 40 years old like I can't do this...I felt badly for him has he felt humiliated." The doctor then goes on to say "he's not capable of making his own decisions." WHO made that determination and HOW was it made? This is vital -- many people with disabilities have been abused by determinations that they cannot make decisions for themselves. Could a SUP method worked instead of a SUB method? Why was that not considered?
I was glad to see in 2.3 and 2.4 about some attention to self-determination, and in 2.5 about keeping the patient informed -- but I would've rather 2.5 had been phrased not with the "importance" of keeping the patient informed, but the "right" of the person to be informed, and for the right of their autonomy to be maximized -- as is embedded in the CRPD.
The article is aimed at clinicians so I see why it is written with that focus, but I feel the article should also take on more the perspective of the patient and their rights.
This is a very important topic - but I can see the disability community reacting to it with lots of red flags. I strongly suggest that the article:
1) Explain the difference between SUB and SUP
2) Acknowledge the rights to SUP, and models of SUP
3) Clearly delineate the conditions under which they consider SUP not possible (my inclination is that they are rare) and the safeguards involved in making the decision to move to SUB
4)Interpret their findings in light of how the two types of decision making can be used.
Reviewer 3 Report
Comments and Suggestions for Authors
This study is a valuable qualitative assessment of how SDMs, clinicians and clients interact. Its primary value is its description of the nuances of this three-way relationship. My main comments have to do with organization:
(1) The abstract sells the article short. The 7 listed items do not capture the richness of the findings, e.g., instead of "Clinicians appreciate hardships and challenges in lives of SDMs and patients" describe what the challenges are; instead of clinicians "have understanding of patient situation and respect for patient autonomy and wishes," describe the tension; instead of clinicians "have a special working relationship with family SDMs," describe what that means. The abstract did not trigger a high degree of interest; I read the article and became very interested. Doing this to the abstract (and in the intro) might make the abstract longer, but some of the text after the list could be incorporated into it.
(2) And the abstract could be shortened in another way. The first two sentences of the abstract talk about the CRPD. Why? The CRPD is mentioned in a couple of places in the article, but that's not really what the article is about. Nor is it primarily about supported decisionmaking v. SDMs. Instead I would not mention the CRPD until the last sentence, and rewrite that last sentence to say something about a comparison of the two approaches (the stuff about the CRPD in the intro could stay, however). Then I could see a whole section in the discussion about pros and cons, based on this research, perhaps starting with the recognition that a lot of important officials in Canada have called for the end of SDM. The focus group comments provide rich ammunition for such a discussion (e.g., for supported decisionmaking: patients feel their autonomy is threatened by SDMs/capacity is not static but fluctuates v. for SDM: very sick individuals can't be counted on to process information/irreversible deterioration could occur because of anosognosia, etc.).
(3) Several other issues about how these decisions are made are briefly waved at in the article but could benefit from more coverage: (a) what happens in emergency situations? (b) what weight are PADs given?; (c) what does incapacity mean (e.g., if a patient clearly understands the risks and benefits but refuses treatment because of a belief that they are not mentally ill, or an exaggerated belief about side effects, or simply disagrees with what consensus medical practice dictates, what weight would/should a clinician/SDM give a refusal? cf. line 216 on p. 5, how do we know a person is "not capable of making decisions"; line 403, p. 9 re finding SDMs incompetent!); (d) all 3 of these issues can be summed up as: when, if ever, do clinicians trump SDMs?
(4) Were there any differences between the inpatient v. outpatient focus groups?;
(5) The finding that public guardians always went along with the doctor is interesting. The article suggests that this is because families are more emotional/worried about harm/more attentive to patient desires. Can this be made more explicit?
(6) Usually the literature review precedes the study description. But I think moving most of that to the discussion works.
I wouldn't insist on any of this, but I think it would make the article even better and more useful.
Round 2
Reviewer 1 Report
Comments and Suggestions for Authors
I want to thank the authors for taking my comments on board and providing a very thoughtful and thorough response to them which I think has greatly clarified and improved the paper.
My only comment is that this sentence should be edited as follows:
"This global movement may have implications for clinicians, who to date are required to conduct clinical care using a capacity-based approach according to domestic law, but who may feel some moral distress in using coercion in their day-to-day work and are aware of the larger global call by some consumer activists to prioritise patient autonomy (Gill et al, 2024)."
I still think that the paper could do with a good proof read with respect to minor grammatical errors, in particular confusion between use of the singular and plural. Sometimes I think it helps to print out the draft and get someone who has never read the paper before to do a good proof-read. It stops the brain from making automatic corrections that aren't actually on the page.
I would like a copy of this paper when it is published as I would like to cite it in a paper I am writing at the moment.
Author Response
Please see attachment and revised paper.

Reviewer 2 Report
Comments and Suggestions for Authors
I thank the authors for their full and thoughtful response to my comments. I think that now someone coming to this article can see better the relation between supported and substitute decision making and the rights of the patient.
Author Response

(The authors gave the same response as above.)
